# Follow-Up of Patients Who Achieved Sustained Virologic Response after Interferon-Free Treatment against Hepatitis C Virus: Focus on Older Patients

**DOI:** 10.3390/medicina57080761

**Published:** 2021-07-27

**Authors:** Kazushige Nirei, Tatsuo Kanda, Ryota Masuzaki, Taku Mizutani, Mitsuhiko Moriyama

**Affiliations:** Department of Medicine, Division of Gastroenterology and Hepatology, Nihon University School of Medicine, 30-1 Oyaguchi-Kamicho, Itabashi-ku, Tokyo 173-8610, Japan; kanda.tatsuo@nihon-u.ac.jp (T.K.); masuzaki.ryota@nihon-u.ac.jp (R.M.); mizutani.taku@nihon-u.ac.jp (T.M.); moriyama.mitsuhiko@nihon-u.ac.jp (M.M.)

**Keywords:** 75 years or older, direct-acting antiviral agent, hepatitis C virus, prognostic factors

## Abstract

*Background and Objectives*: Direct-acting antiviral agents (DAAs) have improved sustained virologic response (SVR) rates in patients with chronic hepatitis C virus (HCV) infection. Our aim was to elucidate the occurrence of hepatocellular carcinoma (HCC) and to compare the outcomes of patients aged 75 years or older (older group) with those of patients younger than 75 years (younger group) after SVR. *Materials and Methods*: Among 441 patients treated with interferon-free DAA combinations, a total of 409 SVR patients were analyzed. We compared the two age groups in terms of HCC incidence and mortality rates. *Results*: Older and younger groups consisted of 68 and 341 patients, respectively. Occurrence of HCC after SVR did not differ between the two groups of patients with a history of HCC. Occurrence of HCC after SVR was observed more in younger patients without a history of HCC (*p* < 0.01). Although older patients without a history of HCC had a higher mortality rate (*p* < 0.01), their causes of death were not associated with liver diseases. Among younger patients without a history of HCC, none died. *Conclusions*: After SVR, liver disease may not be a prognostic factor in older HCV patients without a history of HCC.

## 1. Introduction

The administration of an interferon-free combination of direct-acting antiviral agents (DAAs) with or without ribavirin now allows us to achieve higher sustained virologic response (SVR) rates in hepatitis C virus (HCV) infected patients with fewer side effects than prior interferon-including regimens [1,2]. The treatment options for HCV have also been expanded [3,4,5].

In Japanese phase 3 trials, HCV genotype 1b patients treated for 12 weeks with a combination of the HCV NS5A inhibitor ledipasvir and the NS5B inhibitor sofosbuvir showed an SVR rate of 100% [6]. Additionally, SVR rates of 97–98% were obtained in patients with HCV genotype 2 receiving a 12-week combination of sofosbuvir and ribavirin [7,8]. Thus, regimens that include sofosbuvir could achieve higher SVR rates in Japan [6,7,8,9].

Other interferon-free regimens have shown higher SVR rates than regimens with interferon in Japan [9,10,11]. A 24-week combination of HCV NS3/4A inhibitor asunaprevir plus the NS5A inhibitor daclatasvir led to an 85.2% SVR rate in HCV genotype 1 patients [9]; a 12-week combination of the HCV NS5A inhibitor ombitasvir plus the NS3/4A inhibitor paritaprevir/ritonavir led to SVR rates of 88.9–100% in HCV genotype 1 patients [10]; a 12-week combination of the HCV NS5A inhibitor elbasvir plus the NS3/4A inhibitor grazoprevir led to a 99% SVR rate in HCV genotype 1 patients [11]; and an 8-week or 12-week combination of the HCV NS3/4A inhibitor glecaprevir plus the NS5A inhibitor pibrentasvir led to a 98% SVR rate in HCV pan-genotype patients [11].

We previously reported on the efficacy and outcomes of Japanese patients with genotype 2a or 2b HCV infection treated with sofosbuvir/ribavirin combination therapy in actual clinical settings for 12 weeks, with a mean follow-up period of 2.7 ± 0.8 years [12]. The SVR rate at week 24 (SVR24) was 96.2%. Furthermore, among patients with and without a history of hepatocellular carcinoma (HCC), respectively, 50% and 1.3% developed HCC after SVR24 [12]. As HCV is a major cause of HCC [9], it is important to investigate whether the occurrence of HCC decreases or not after SVR in DAA recipients.

Additionally, two studies with large cohorts obtained a 98% SVR rate with DAAs, regardless of age. SVR after treatment with DAA reduced the progression to severe cirrhosis and the occurrence of HCC [13,14]. In Japan, the physical function of older patients was improved for several years. Therefore, the definition of old age was changed from 65 years to 75 years and older [15]. Previous reports showed that age is not related to treatment effects, with an SVR rate of 98.8% for those aged younger than 75 and 97.5% for those aged older than 75 years. The incidence of adverse events did not increase with age (2.9% in the younger group and 3.0% in the older group) [3].

However, to our knowledge, no studies have focused on HCV patients aged 75 years or older regarding long-term clinical outcomes. In the present study, we compared long-term clinical outcomes in SVR24 patients treated with interferon-free DAA aged 75 years or older and younger than 75 years. We especially focused on the incidence of HCC and showed that regular screening for HCC is needed in HCV patients after SVR.

## 2. Materials and Methods

### 2.1. Patients

All subjects received interferon-free DAA treatment from 1 October 2014 through 26 July 2019. In total, 441 HCV patients were treated with interferon-free combinations of DAAs at Nihon University School of Medicine Itabashi Hospital, Tokyo, Japan. Two patients were excluded because of coinfection with hepatitis B virus. No patients had human immunodeficiency virus infection. None of the patients were current intravenous drug abusers, and none were pregnant. All had chronic hepatitis or Child–Pugh A cirrhosis. Patients with Child‒Pugh B or C cirrhosis were excluded from the present study. In the present analysis, only those who achieved SVR with the initial treatment were considered, meaning that we excluded patients receiving multiple DAA regimens. Finally, we included patients who were followed for at least 0.5 years after SVR24 (mean follow-up period, 3.69 ± 1.23 years; Figure 1). Patients with a history of the use of interferon were also excluded.

### 2.2. Drug Regimens Used for SVR24 in the Present Study

As shown in Table 1, SVR24 was achieved in 409 patients with a 24-week combination of asunaprevir plus daclatasvir, a 12-week combination of sofosbuvir plus ribavirin, a 12-week combination of sofosbuvir plus ledipasvir, a 12-week combination of ombitasvir plus paritaprevir and ritonavir, a 12-week combination of elbasvir plus grazoprevir, or an 8- or 12-week combination of glecaprevir plus pibrentasvir [6,7,8,9,10,11,12].

### 2.3. Laboratory Tests

Laboratory tests were performed at least every 4 weeks before and after the interferon-free DAA therapy. Before starting the DAAs, we measured the levels of serum aspartate aminotransferase (AST) and alanine aminotransferase (ALT) and determined the serum platelet count and HCV genotypes. All patients were positive for the HCV antibody (by second-generation ELISA; Abbott, Tokyo, Japan). We confirmed HCV positivity and measured the HCV RNA levels in blood samples using the Cobas TaqMan HCV method (Roche Diagnostics, Meylan, France). HCV genotypes were examined by a previously described method [12,16]. Undetectable HCV RNA at 24 weeks post-treatment was defined as SVR (SVR24) in the present study.

### 2.4. Diagnosis of Cirrhosis and HCC

The diagnosis of cirrhosis was based on an FIB-4 index of at least 3.25, according to a previously published report [17]. Before treatment, the absence of malignant diseases was confirmed by blood biochemical, ultrasound, chest X-ray, and endoscopic examinations. Malignancy was an exclusion criterion. By the end of the treatment, the patients had undergone abdominal ultrasonography every 3 to 6 months and abdominal dynamic computed tomography or magnetic resonance imaging every 6 to 12 months to detect HCC.

This study was approved by the Hospital Institutional Review Board of the Nihon University School of Medicine Itabashi (RK-181009-4) and conformed to the ethical guidelines of the Declaration of Helsinki. Participation in the study was posted at the website of our institution, and informed consent was obtained from all patients.

### 2.5. Statistical Analysis

Patients were divided into two groups: patients aged 75 years or older (older group) and patients younger than 75 years (younger group). Differences between the 2 groups were analyzed using the Mann‒Whitney U-test, chi-squared test, and Wilcoxon signed-rank test. The cumulative incidence of HCC and survival rates between the 2 groups were compared using Gray’s test. All statistical analyses were performed using the EZR (Easy R) software. EZR is freely available on the website (http://www.jichi.ac.jp/saitama-sct/SaitamaHP.files/statmed.html, accessed date: 27 July 2021), which is a modified version of R commander, designed to add frequently used statistical functions in biostatistics [18].

## 3. Results

### 3.1. Comparison of Pretreatment Factors for Patients Aged 75 Years or Older and Younger Than 75 Years

Patient characteristics are shown in Table 2. In a comparison of background features between the two groups, we observed a greater proportion of female patients (*p* < 0.01), lower hemoglobin levels (*p* < 0.01), lower platelet counts (*p* = 0.02), lower estimated glomerular filtration rates (eGFRs) (*p* < 0.01), a higher distribution of HCV genotype 1b (*p* < 0.01), and a history of HCC (*p* = 0.02) in the older group (Table 2).

### 3.2. Comparison of Post-Treatment Factors for Patients Aged 75 Years or Older and Younger Than 75 Years

We compared ALT levels, AST levels, platelet counts, and total bilirubin levels before and after SVR24 for the older and younger groups using the Wilcoxon signed-rank test (Table 3). The ALT and AST levels of the two groups were significantly lower after SVR24 (*p* < 0.01). After SVR24, platelet counts were significantly lower in the younger group and showed a decreasing trend in the older group, but the difference was not statistically significant (Table 3).

### 3.3. Occurrence of HCC in Patients with or without a History of HCC

We compared the patients in terms of their HCC history (Figure 2). Among patients with a history of HCC, the cumulative probability of HCC occurrence in those aged younger than 75 years was 0% at 1 year, 17.65% at 2 years, 25.13% at 3 years, and 61.72% at 4 years. Among patients with a history of HCC, the cumulative probability of HCC occurrence in those aged 75 years or older was 0% at 1 year, 20.0% at 2 years, 20.0% at 3 years, and 31.4% at 4 years. Among those with a history of HCC, 3 of 10 patients in the older group and 4 of 18 patients in the younger group developed HCC during the follow-up periods. Thus, the HCC incidence rates of the two age groups were not significantly different (*p* = 0.90) (Figure 2A).

Among patients without a history of HCC, the cumulative probability of HCC occurrence in those aged younger than 75 years was 0% at 1 year, 0.3% at 2 years, 1.12% at 3 years and 3.05% at 4 years. Among patients without a history of HCC, the cumulative probability of HCC occurrence in those aged 75 years or older was 0% (Figure 2B). Among those without a history of HCC, 0 of the 58 patients in the older group and 12 of 323 patients in the younger group developed HCC during the follow up periods (*p* < 0.01). Of note, among those without a history of HCC, no one in the group aged 75 years or older developed HCC (Figure 2B).

### 3.4. Clinical Outcomes of Patients with or without a History of HCC

We also compared the patients in terms of their HCC history. Among those with a history of HCC, in the group younger than 75 years, a total of three patients died during the follow up-periods. The cause of death was HCC in two patients and unknown in one patient. The times of death were 0.62, 1.95, and 3.10 years, respectively. Survival probability was 78.4% at 4 years. In the group 75 years or older, a total of two patients died during the follow-up periods. The cause of death was HCC in one patient and dehydration in one patient. The times of death were 3.96 and 3.10 years, respectively. Survival probability was 76.2% at 4 years. Among those with a history of HCC, age had no impact on their survival rates in the present study (*p* = 0.74) (Figure 3A).

Among those without a history of HCC, in the group younger than 75 years, no patients died. In the group 75 years or older, three patients died. The causes of death were chronic respiratory failure (0.51 years), pancreatic cancer (2.65 years), and sudden death (3.27 years) (*p* < 0.01) (Figure 3B). Thus, among those without a history of HCC, no patients died due to liver diseases. Survival probability was 93.4% at 4 years.

## 4. Discussion

Similarly to other studies, we found that ALT and AST levels improved regardless of age [8]. Regarding the HCV genotypes in the younger group, 4, 214, 74, 43, and 3 had HCV genotypes 1a, 1b, 2a, 2b, and 3a, respectively. Although the HCV subgenotype was undetermined in three patients, these were HCV genotype 2. In the older group, 58, 7, and 3 had HCV genotypes 1b, 2a, and 2b, respectively. Of interest, the prevalence of HCV genotype 2 in the younger group was higher than that in the older group (chi-squared test; *p* < 0.05).

As for the optimal age for DAA treatment compared with interferon, it was reported that among 507 patients treated with sofosbuvir plus ledipasvir, only 0.8% discontinued the treatment due to adverse events. They compared the SVR rates among subgroups of the entire cohort aged younger than 65 years, 65 to 75 years, and 75 years and older and found no differences [5]. The treatment effects did not differ between the age groups, and the incidence of adverse events did not increase due to the patient’s age, supporting the results of the present study.

Older people have many other health complications besides liver diseases. Taking various medicines is also considered an important factor in DAA treatment [3]. In our study, the SVR rate reached 68/75 (90.67%) in the older group. Regarding adverse events, 2 patients treated with sofosbuvir-based regimens had ventricular tachycardia that was resolved by discontinuation [19]. One patient discontinued DAA treatment due to the occurrence of cerebral hemorrhage during the combination treatment of asunaprevir plus daclatasvir. More careful attention should be paid to drug–drug interactions in DAA treatment for older HCV patients, although clinicians could access various DAA regimens as personalized medicine.

Another study compared carcinogenesis rates after DAA and interferon treatment and showed no significant differences [20]. Other reports have stated that DAAs reduce the risk of HCC and decompensated cirrhosis, thereby reducing all-cause mortality [13,21]. In Japan, patients with chronic HCV infection tend to be older, thus may be at increased risk of developing HCC [22]. In the present study, we examined the recurrence and initial onset of HCC after DAA treatment. No significant differences were observed in the HCC occurrence rates between two groups of patients with a history of HCC.

Our study supports a previous report showing that the occurrence of HCC was more often observed in patients with a history of HCC [23]. There is a certain report that DAA administration after HCC treatment was reported to exacerbate HCC [24]. Another report indicated that the carcinogenesis rate is unaffected by DAAs [25]. Other reports indicated that DAA treatment improved outcomes after HCC treatment [13,20,26,27,28,29]. Of note, we found that younger patients without a history of HCC seemed to have higher occurrence of HCC, although the number of patients was relatively small. It is possible that HCV patients who do not develop HCC by the age of 75 may have a lower hepatocarcinogenic status. Another limitation of this study is that the number of older patients was too small.

Backus et al. reported that DAA-induced SVR decreased mortality and HCC development in 15,059 HCV patients with progressive liver disease, as defined by a FIB-4 index >3.25 [30]. DAA treatment reportedly reduced the number of deaths of patients with advanced liver disease [30]. We compared the post-treatment factors between patients aged 75 years or older and those younger than 75 years. In both groups, ALT and AST levels were significantly lower at SVR24 than at baseline (Table 3). Platelet counts were elevated in the younger group and showed an increasing trend in the older group, but the difference was not statistically significant (Table 3). Among the elements of the FIB-4 index, ALT levels and platelet counts were involved. No cases of ascites or other exacerbations occurred during the course of treatment. In the present study, no patients developed decompensated cirrhosis.

Carrat et al. investigated 7344 HCV patients who were treated with DAAs and compared them to those who were not [13]. They reported that DAA administration improved HCC incidence and mortality rates. The rate of transition to cirrhosis was not related [13]. Our results also support their report.

The risk factors for HCC in patients with SVR receiving interferon treatment are age, male gender, progression of liver fibrosis and steatosis, low serum cholesterol level, hyperglycemia, and higher ALT and alpha-fetoprotein levels, although we did not measure alpha-fetoprotein levels in all patients. Previous studies reported that older age is a risk factor for the occurrence of HCC in HCV patients treated with DAAs [11,21]. In the present study, we did not identify any impact of older age on the occurrence of HCC after SVR in HCV patients with or without a history of HCC. This point is the novelty of our study.

Among those with a history of HCC, does DAA administration after HCC treatment increase the overall survival of HCC patients? Although the number of patients in the present study decreased at 5 years, among patients with a history of HCC, the cumulative probability of HCC occurrence in those aged younger than 75 years or those aged 75 years or older was 25.13% or 31.4% at 5 years, respectively. Survival probability at 5 years in those aged younger than 75 years and those aged 75 years or older was 78.7% and 76.2%, respectively. Shiratori et al. reported that survival rates at 5 years in patients treated with or without interferon after tumor ablation for HCC were 68% or 48%, respectively [31]. In Japan, 5-year relative survival rates after the diagnosis of HCC were 38% and 38.4%, respectively, in men and women [32]. Therefore, DAA administration after HCC treatment seems to improve the overall survival of HCC patients.

It is important to perform HCC surveillance using imaging modalities, such as abdominal ultrasonography, computed tomography, or magnetic resonance imaging, every 6 months, combined with the use of tumor markers such as alpha-fetoprotein, lens culinaris agglutinin-reactive alpha-fetoprotein isoform (AFP-L3), and/or des-γ-carboxy prothrombin (DCP) [11]. This type of surveillance seems to represent an improvement in the prognosis of liver disease-related death.

## 5. Conclusions

DAA treatment did not increase the occurrence of HCC, especially in HCV patients aged 75 years or older without a history of HCC. Among patients with a history of HCC, the prognosis was not associated with age. However, the prognosis factors for these patients were not related to liver disease. After treatment with DAA against HCV, older age is not always a factor in HCC development. We reconfirmed that close surveillance and monitoring for HCC occurrence are required in HCV patients who undergo DAA treatment and achieve SVR.

## Figures and Tables

**Figure 1 medicina-57-00761-f001:**
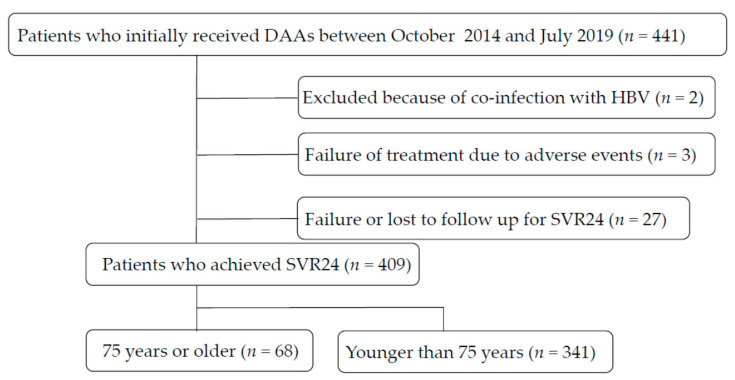
Flowchart showing patients enrolled in this study. DAAs, direct-acting antiviral agents; HBV, hepatitis B virus; SVR, sustained virologic response.

**Figure 2 medicina-57-00761-f002:**
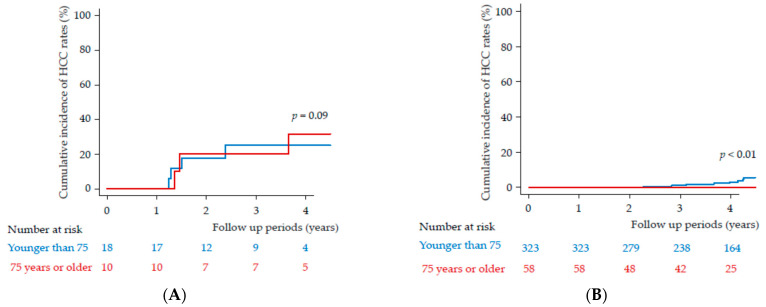
Occurrence of hepatocellular carcinoma (HCC) in patients (**A**) with or **(B)** without a history of HCC. Red line indicates 75 years or older; blue line indicates younger than 75 years.

**Figure 3 medicina-57-00761-f003:**
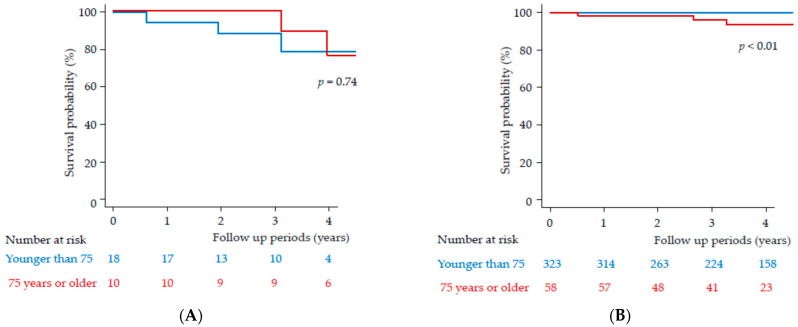
Survival probability in patients (**A**) with or (**B**) without a history of hepatocellular carcinoma. Red line indicates 75 years or older; blue line indicates younger than 75 years.

**Table 1 medicina-57-00761-t001:** Interferon-free combinations of direct-acting antiviral agents for SVR24 patients in this study.

Combination Regimens	Total Patients (*n*)	75 Years or Older (*n*)	Younger Than 75 Years (*n*)
Asunaprevir plus daclatasvir	62	15	47
Sofosbuvir plus ribavirin	95	8	87
Sofosbuvir plus ledipasvir	146	22	124
Ombitasvir plus paritaprevir/ritonavir	19	5	14
Glecaprevir plus pibrentasvir	82	16	66
Elbasvir plus grazoprevir	5	2	3
Total SVR patients	409	68	341

SVR, sustained virologic response. References [6,7,8,9,10,11,12].

**Table 2 medicina-57-00761-t002:** Background features of the two age groups: patients 75 years or older and younger than 75 years.

Items	Total	75 Years or Older	Younger Than 75 Years	*p*-Value
Number	409	68	341	
Age (years)	62.53 ± 11.56	78.69 ± 2.94	59.31 ± 9.80	<0.01
Male/Female	185/224	19/49	166/175	<0.01
AST (IU/L)	51.95 ± 38.60	49.02 ± 28.09	52.20 ± 40.37	0.34
ALT (IU/L)	53.02 ± 47.11	39.00 ± 21.70	54.82 ± 50.32	0.06
Hemoglobin (g/dL)	13.88 ± 6.74	12.63 ± 1.19	14.14 ± 7.04	<0.01
Platelet count (×10⁴/mm^3^)	17.75 ± 7.15	16.02 ± 4.69	18.09 ± 8.05	0.02
eGFR (mL/min/1.73 m^2^)	74.79 ± 18.97	63.94 ± 15.47	77.31 ± 18.45	<0.01
Prothrombin time (%)	93.81 ± 13.87	95.51 ± 6.85	93.60 ± 14.38	0.09
HCV RNA (log IU/mL)	5.92 ± 0.91	5.94 ± 0.81	5.92 ± 0.93	0.80
HCV genotypes (1/2/others)	276/130/3	58/10/0	218/120/3	<0.01
Diabetes mellitus (%)	66	10 (13.69%)	56 (13.74%)	0.85
Liver cirrhosis (%)	138 (33.74%)	40 (58.82%)	98 (28.73%)	<0.01
History of interferon (%)	80 (19.56%)	16 (23.52%)	64 (19.06%)	0.62
History of HCC (%)	28 (0.01%)	10 (18.8%)	18 (5.5%)	0.01

AST, aspartate aminotransferase; ALT, alanine aminotransferase; eGFR, estimated glomerular filtration rate; HCV, hepatitis C virus; HCC, hepatocellular carcinoma.

**Table 3 medicina-57-00761-t003:** Differences in laboratory data between pre-treatment and post-SVR24 in two age groups.

75 Years or Older (*n* = 68)	Pre-Treatment	Post-SVR24	*p*-Value
ALT (IU/L)	39.00 ± 21.70	16.07 ± 6.87	<0.01
AST (IU/L)	49.02 ± 28.09	26.02 ± 7.38	<0.01
Platelet count (×10^4^/mm^3^)	16.02 ± 4.69	16.76 ± 5.30	0.06
Total bilirubin (mg/dL)	0.59 ± 0.22	0.60 ± 0.24	0.69
**Younger Than 75 Years (*n* = 341)**	**Pre-Treatment**	**Post-SVR24**	***p*-Value**
ALT (IU/L)	54.82 ± 50.32	28.38 ± 170.95	<0.01
AST (IU/L)	52.20 ± 40.37	25.54 ± 17.76	<0.01
Platelet count (×10^4^/mm^3^)	18.09 ± 8.05	18.96 ± 7.43	<0.01
Total bilirubin (mg/dL)	0.69 ± 0.51	0.84 ± 2.19	0.51

Data are expressed as mean ± standard deviation. AST, aspartate aminotransferase; ALT, alanine aminotransferase.

## Data Availability

The study did not report any data. The work was conducted at the Division of Gastroenterology and Hepatology, Department of Medicine, Nihon University School of Medicine, Tokyo, Japan.

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
