# Peer review of "Follow-Up of Patients Who Achieved Sustained Virologic Response after Interferon-Free Treatment against Hepatitis C Virus: Focus on Older Patients"

_medicina, 2021, doi:10.3390/medicina57080761_

Round 1

Reviewer 1 Report

interesting work 

Author Response

To Reviewer 1:

Thank you for your encouraging comment. Accordingly, we also made corrections of grammatical and minor spell errors with red color font.

Reviewer 2 Report

The article ‘’ Follow-up of patients who achieved sustained virologic 2 response after interferon-free treatment against hepatitis C virus: focus on older patients’’ is a well written article by Kazushige et al. In this article the authors have analyzed the Direct-acting antiviral agents (DAAs) in patients with chronic hepatitis C virus (HCV) infection. The authors also compared the SVR between young and old patients which is novel and could give insights to understand whether after SVR, liver disease may not be a prognostic factor in older HCV patients without a history of HCC. Overall the article would be a good addition to the journal and its readers and I have few comments and if addressed would improve the article.

  1. The authors should elaborate on the older people population could have many other health complications besides liver diseases. Taking various medicines is also considered an important factor in DAA treatment and hence the authors need to explain if the other medication could impact the SVR. 
  1. The authors should check if DAA administration after HCC treatment could increase the overall survival of HCC patients.
  1. The authors should discuss the off-target side effects of DAA administration.

Author Response

To Reviewer 2:

Thank you for your invaluable comments.

Response to your comment 1 :“The authors should elaborate on the older people population could have many other health complications besides liver diseases. Taking various medicines is also considered an important factor in DAA treatment and hence the authors need to explain if the other medication could impact the SVR.”

Thank you for your valuable comments. We agree with you. We do not always think that the other medication could have impacts on the SVR at the present because clinicians could access various DAA regimens as personalized medicine. We revised our manuscript as follows.

In Discussion section, page 7, lines 200-208,

Older people have many other health complications besides liver diseases. Taking various medicines is also considered an important factor in DAA treatment [3]. In our experience, the SVR rates reached 68/75 (90.67%) in the older group. Regarding adverse events, 2 patients treated with sofosbuvir-based regimens had ventricular tachycardia that were resolved by discontinuation [19]. One patient discontinued DAA treatment due to the occurrence of cerebral hemorrhage during the combination treatment of asunaprevir plus daclatasvir. More careful attention should be paid to drug–drug interactions in DAA treatment for older HCV patients although clinicians could access various DAA regimens as personalized medicine.

Response to your comment 2 :“The authors should check if DAA administration after HCC treatment could increase the overall survival of HCC patients.”

Thank you for your valuable comments. We agree with you. We revised our manuscript as follows.

In Discussion section, page 8, lines 249-259,

Among those with a history of HCC, does DAA administration after HCC treatment increase the overall survival of HCC patients? Although the number of patients in the present study decreased at 5 years, among patients with a history of HCC, the cumulative probability of HCC occurrence in those aged younger than 75 years or those aged 75 years or older was 25.13% or 31.4% at five years, respectively. Survival probability at 5 years in those aged younger than 75 years and those aged 75 years or older were 78.7% and 76.2%, respectively. Shiratori at al. reported that survival rates at 5-year in patients treated with or without interferon after tumor ablation for HCC were 68% or 48%, respectively [31]. In Japan, 5-year relative survival rates after the diagnosis of HCC were 38% and 38.4%, respectively, in men and women [32]. Therefore, DAA administration after HCC treatment seem to improve the overall survival of HCC patients.

Response to your comment 3 :“The authors should discuss the off-target side effects of DAA administration.”

Thank you for your valuable comments. We agree with you. We revised our manuscript as follows.

In Discussion section, page 7, lines 200-208,

Older people have many other health complications besides liver diseases. Taking various medicines is also considered an important factor in DAA treatment [3]. In our experience, the SVR rates reached 68/75 (90.67%) in the older group. Regarding adverse events, 2 patients treated with sofosbuvir-based regimens had ventricular tachycardia that were resolved by discontinuation [19]. One patient discontinued DAA treatment due to the occurrence of cerebral hemorrhage during the combination treatment of asunaprevir plus daclatasvir. More careful attention should be paid to drug–drug interactions in DAA treatment for older HCV patients although clinicians could access various DAA regimens as personalized medicine.
